microbiology, cellular biology, ecology

chytrids, rhizoid, hyphae, morphogenesis, plasticity, fungi

**Author for correspondence:**
Michael Cunliffe
e-mail: micnli@mba.ac.uk

# Chytrid rhizoid morphogenesis resembles hyphal development in multicellular fungi and is adaptive to resource availability

Davis Laundon[1,2], Nathan Chrismas[1,3], Glen Wheeler[1] and Michael Cunliffe[1,4]

[1]Marine Biological Association of the UK, The Laboratory, Citadel Hill, Plymouth, UK
[2]School of Environmental Sciences, University of East Anglia, Norwich, UK
[3]School of Geographical Sciences, University of Bristol, Bristol, UK
[4]School of Biological and Marine Sciences, University of Plymouth, Plymouth, UK

MC, 0000-0002-6716-3555

Key to the ecological prominence of fungi is their distinctive cell biology, our understanding of which has been principally based on dikaryan hyphal and yeast forms. The early-diverging Chytridiomycota (chytrids) are ecologically important and a significant component of fungal diversity, yet their cell biology remains poorly understood. Unlike dikaryan hyphae, chytrids typically attach to substrates and feed osmotrophically via anucleate rhizoids. The evolution of fungal hyphae appears to have occurred from rhizoid-bearing lineages and it has been hypothesized that a rhizoid-like structure was the precursor to multicellular hyphae. Here, we show in a unicellular chytrid, *Rhizoclosmatium globosum*, that rhizoid development exhibits striking similarities with dikaryan hyphae and is adaptive to resource availability. Rhizoid morphogenesis exhibits analogous patterns to hyphal growth and is controlled by β-glucan-dependent cell wall synthesis and actin polymerization. Chytrid rhizoids growing from individual cells also demonstrate adaptive morphological plasticity in response to resource availability, developing a searching phenotype when carbon starved and spatial differentiation when interacting with particulate organic matter. We demonstrate that the adaptive cell biology and associated developmental plasticity considered characteristic of hyphal fungi are shared more widely across the Kingdom Fungi and therefore could be conserved from their most recent common ancestor.

## 1. Introduction

Hyphae are polarized, elongating and bifurcating cellular structures that many fungi use to forage and feed (figure 1*a* and *b*). The phylum Chytridiomycota (chytrids) diverged from other fungal lineages approximately 750 Mya and, with the Blastocladiomycota, formed a critical evolutionary transition in the Kingdom Fungi dedicated to osmotrophy and the establishment of the chitin-containing cell wall [2]. Chytrids produce filamentous hyphae-like, anucleate structures called rhizoids (figure 1*a–c*) [3], which are important in their ecological functions, in terms of both attachment to substrates and osmotrophic feeding [2]. 407-million-year-old fossils from the Devonian Rhynie Chert deposit show chytrids in freshwater aquatic ecosystems physically interacting with substrates via rhizoids in a comparative mode to extant taxa [4]. Yet surprisingly, given the importance of rhizoids in both contemporary and paleo-chytrid ecology, there remains a limited understanding of chytrid rhizoid biology, including possible similarities with functionally analogous hyphae in other fungi and the potential for substrate-dependent adaptations.

Character mapping of the presence of cellular growth plans against established phylogenies reveals the multicellular hyphal form to be a derived condition, whereas rhizoid feeding structures are the basal condition within the true fungi

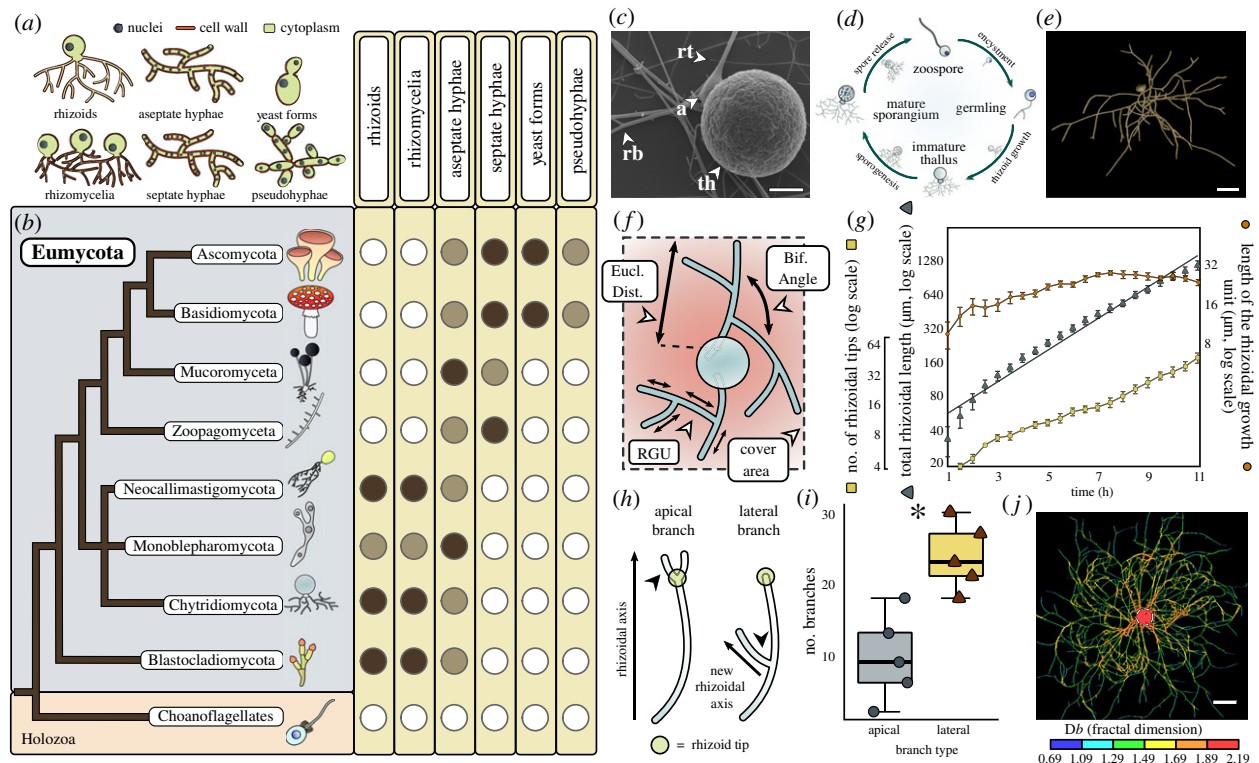

**Figure 1.** Rhizoids are the basal feeding condition within the fungal kingdom and their morphogenesis is similar to hyphal development. (*a–b*) Correlating the major feeding types in fungi (*a*) to phylogeny (*b*) shows rhizoids to be the basal feeding condition in the true fungi (Eumycota). White circles indicate absence of a growth plan in a taxon and dark circles indicate widespread presence. Faded circles indicate a growth plan is present within a taxon, but not widespread. Rhizoid systems bear a single thallus, whereas rhizomycelia are defined as rhizoid systems with multiple thalli. Tree based on the phylogeny outlined in [1]. (*c*) *R. globosum* displays a typical chytrid cell plan. Shown is the thallus (th) anchored to the substrate by threadlike rhizoids. Rhizoids emanate from a swelling termed the apophysis (a). Also shown are rhizoid bifurcations (rb) and tips (rt). Scale bar = 5 μm. (*d*) *R. globosum* exhibits an archetypal chytrid life cycle. From the mature zoosporangium, multiple motile zoospores are released that swim freely for a brief period (less than 1–2 h) before encysting into germlings (i.e. losing the single flagellum and developing a chitin-containing cell wall). From the subsequent extending germtube develops the rhizoid. After a period of growth, the sporangium becomes the multinucleate zoosporangium during sporogenesis, from which the next generation of spores is released. (*e*) Chytrid rhizoids were reconstructed using the neuron tracing workflow outlined in electronic supplementary material, figure S3. Example of a three-dimensional reconstructed *R. globosum* rhizoid system taken from a 10 h time series. Scale bar = 20 μm. (*f*) This study used morphometric parameters developed from neuron biology to described rhizoid development in chytrids. Shown are the Euclidean distance (Eucl. Dist.), bifurcation angle (Bif. Angle), rhizoid growth unit (RGU) and cover area. Full morphometric glossary is presented in electronic supplementary material, figure S5. (*g*) Rhizoid growth trajectories for four-dimensional confocal time series (*n* = 5, mean ± s.e.m.) of rhizoidal growth unit, total length and number of tips. (*h*) Apical and lateral branches occur in chytrid rhizoids. Apical branching occurs when a branch is formed at the rhizoid tip parallel to the established rhizoidal axis. Lateral branching occurs when a branch is formed distally to the rhizoidal tip, establishing a new rhizoidal axis. (*i*) Four-dimensional confocal imaging (*n* = 5, mean ± s.e.m.) revealed that lateral branching dominates over apical branching *$p < 0.05$. (*j*) Fractal analysis of chytrid rhizoid systems shows a decrease in fractal dimension (D*b*) towards the growing edge. Thallus demarked by dashed circle. Scale bar = 50 μm.

(Eumycota) (figure 1*a* and *b*). Aseptate hyphae represent an intermediary condition and are not typically the dominant cell type in either unicellular or multicellular fungi. Hyphal cell types are sometimes observed outside of the Eumycota, such as within the Oomycota; however, the origin of fungal hyphae within the Eumycota was independent [5,6] and has not been reported in their closest relatives the Holozoans (animals, choanoflagellates, etc.). Comparative genomics has indicated that hyphae originated within the rhizoid-bearing Chytridiomycota–Blastocladiomycota–Zoopagomycota nodes of the fungal tree [6], which is supported by fossil Blastocladiomycota and extant Monoblepharidomycetes having hyphae [5,7]. This has led to the proposition that rhizoids, or rhizoid-like structures, were the evolutionary precursors of fungal hyphae [5,6,8]; however, investigation into such hypotheses have been hindered by a relative lack of understanding of rhizoid developmental biology.

Chytrids are important aquatic fungi [9], feeding on a range of physically complex heterologous substrates, including algal cells [10], amphibian epidermises [11] and recalcitrant particulate organic matter (POM) such as chitin and pollen [12]. Appreciation for the ecological importance of chytrids as parasites, pathogens and saprotrophs in aquatic ecosystems is greatly expanding. For example, chytrids are well-established plankton parasites [10], responsible for the global-scale amphibian panzootic [13], and have recently emerged as important components of the marine mycobiome [9]. The chytrid rhizoid is critical in all ecological functions because it is the physical interface between the fungus and substrate or host, yet there remains a limited understanding of rhizoid functional biology in terms of substrate interaction.

*Rhizoclosmatium globosum* is a widespread aquatic saprotroph that is characteristically associated with chitin-rich insect exuviae and has an archetypal chytrid cell plan (figure 1*c*) and life cycle (figure 1*d*) [14]. With an available sequenced genome [15], easy laboratory culture and amenability to live cell imaging (this study), *R. globosum* JEL800 represents a promising model organism to investigate the cell and developmental biology of aquatic rhizoid-bearing, early-diverging fungi. To

study the developing rhizoid system for morphometric analyses, we established a live cell three-/four-dimensional confocal microscopy approach in combination with the application of neuron tracing software to three-dimensional reconstruct developing cells (figure 1e; electronic supplementary material, figures S3 and S4). We were subsequently able to generate a series of cell morphometrics to describe and quantify rhizoid development (figure 1f; electronic supplementary material, figure S5) under a range of experimental conditions with the aims of identifying potential similarities with hyphae in dikaryan fungi in terms of geometric organization, morphogenesis and underlying cellular control mechanisms. In addition, we set out to characterize substrate-dependent adaptations particularly in the ecological context of aquatic POM utilization.

## 2. Material and methods

Detailed materials and methods are provided as electronic supplementary material.

### (a) Rhizoid tracing and reconstruction

Chytrid plasma membranes were labelled with 8.18 μM FM 1–43 and imaged using a Zeiss LSM 510 Meta confocal laser scanning microscope (Carl Zeiss). Z-stacks of rhizoids were imported into the neuron reconstruction software NeuronStudio [16,17] and semi-automatically traced with the 'Build Neurite' function. Rhizoids were morphometrically quantified using the btmorph2 library [18] run with Python 3.6.5 implemented in Jupyter Notebook 4.4.0. For visualization, reconstructed rhizoids were imported into Blender (2.79), smoothed using automatic default parameters and rendered for display only.

### (b) Chemical characterization of the rhizoid

To label F-actin and the cell wall throughout the rhizoid system, cells were fixed for 1 h in 4% formaldehyde in 1×PBS (phosphate-buffered saline) and stained with 1 : 50 rhodamine phalloidin in PEM (100 mM PIPES (piperazine-N,N′-bis(2-ethanesulfonic acid)) buffer at pH 6.9, 1 mM EGTA (ethylene glycol tetraacetic acid), and 0.1 mM MgSO$_4$) for 30 min, then with 5 μg ml$^{-1}$ Texas Red-conjugated wheat germ agglutinin (WGA) in PEM for 30 min.

### (c) Chemical inhibition of rhizoid growth

Caspofungin diacetate (working concentration 1–50 μM) was used to inhibit cell wall β-glucan synthesis and cytochalasin B (working concentration 0.1–10 μM) was used to inhibit actin filament formation. Cells were incubated for 6 h, which was found to be sufficient to observe phenotypic variation.

### (d) β-glucan quantification

R. globosum was processed for β-glucans using a commercial β-glucan assay (Yeast & Mushroom) (K-YBGL, Megazyme) following the manufacturer's protocol. Briefly, samples were processed by acid hydrolysis then enzymatic break-down and β-glucans were quantified spectrophotometrically with a CLARIOstar Plus microplate reader (BMG Labtech), relative to a negative control and positive β-glucan standard. A sample of shop-bought baker's yeast was used as an additional positive control.

### (e) Identification of putative glucan synthase genes

All glycosyl transferase group 2 (GT2) domain-containing proteins within the R. globosum genome were identified using the JGI MycoCosm online portal. GT2 functional domains were identified using DELTA-BLAST [19] and aligned with MAFFT [20]. Maximum-likelihood phylogenies were calculated with RAxML [19] using the BLOSUM62 matrix and 100 bootstrap replicates.

### (f) Carbon starvation and growth on chitin beads

For carbon starvation experiments, R. globosum cells were grown in either carbon-free Bold's Basal Medium supplemented with 1.89 mM ammonium sulfate and 500 μl l$^{-1}$ f/2 vitamin solution [21] (BBM) or BBM with 10 mM N-acetyl-D-glucosamine as a carbon source. To investigate growth on POM, chitin microbeads (New England Biolabs) were suspended in carbon-free BBM at a working concentration of 1 : 1000 stock concentration. To understand rhizoid development in a starved cell that had encountered a chitin microbead, we imaged cells that contacted a chitin microbead following development along the glass bottom of the dish.

## 3. Results

### (a) Chytrid rhizoid morphogenesis and development

During rhizoid development, we observed a continuous increase in rhizoid length ($110.8 \pm 24.4$ μm h$^{-1}$) ($n = 5$, ± s.d.) and the number of rhizoid tips ($4.6 \pm 1.2$ tips h$^{-1}$) (figure 1g; electronic supplementary material, table S1, movies S1–S5), with an increase in the total cell surface area ($21.1 \pm 5.2$ μm$^2$ h$^{-1}$), rhizoid bifurcations ($4.2 \pm 1.0$ bifurcations h$^{-1}$), cover area ($2,235 \pm 170.8$ μm$^2$ h$^{-1}$) and maximum Euclidean distance ($5.4 \pm 0.1$ μm h$^{-1}$) (electronic supplementary material, figure S6). The hyphal growth unit (HGU) has been used previously to describe hyphal development in dikaryan fungi and is defined as the distance between two hyphal bifurcations [22]. Adapting this metric for the chytrid rhizoid, the rhizoidal growth unit (RGU) (i.e. the distance between two rhizoid bifurcations; figure 1f) increased continuously during the first 6 h of the development period (i.e. cells became relatively less branched) before stabilizing during the later phase of growth (figure 1g). The local rhizoid bifurcation angle remained consistent at $81.4° \pm 6.3$ after approximately 2 h (electronic supplementary material, figure S6), and lateral branching was more frequent than apical branching during rhizoid development (figure 1h and i). Fractal analysis (fractal dimension = D$b$) of 24 h grown cells showed that rhizoids approximate a two-dimensional biological fractal (mean D$b = 1.51 \pm 0.24$), with rhizoids relatively more fractal at the centre of the cell (max D$b = 1.69$–$2.19$) and less fractal towards the growing periphery (min D$b = 0.69$–$1.49$) (figure 1j; electronic supplementary material, figure S7).

### (b) Cell wall and actin dynamics are linked to rhizoid branching

The cell wall and actin structures were present throughout the chytrid rhizoid (figure 2a). Putative actin cables ran through the rhizoid system, punctuated by actin patches. Inhibition of cell wall β-1,3-glucan synthesis and actin proliferation with caspofungin and cytochalasin B, respectively, induced a concentration-dependent decrease in the RGU, with the development of atypical hyperbranched rhizoids (figure 2b–d; electronic supplementary material, table S2, movies S6–S7). As with Batrachochytrium dendrobatidis [23,24], we confirmed that R. globosum JEL800 lacks an apparent β-1,3-glucan synthase FKS1 gene homologue (electronic supplementary material, table S3). However, the quantification of glucans in

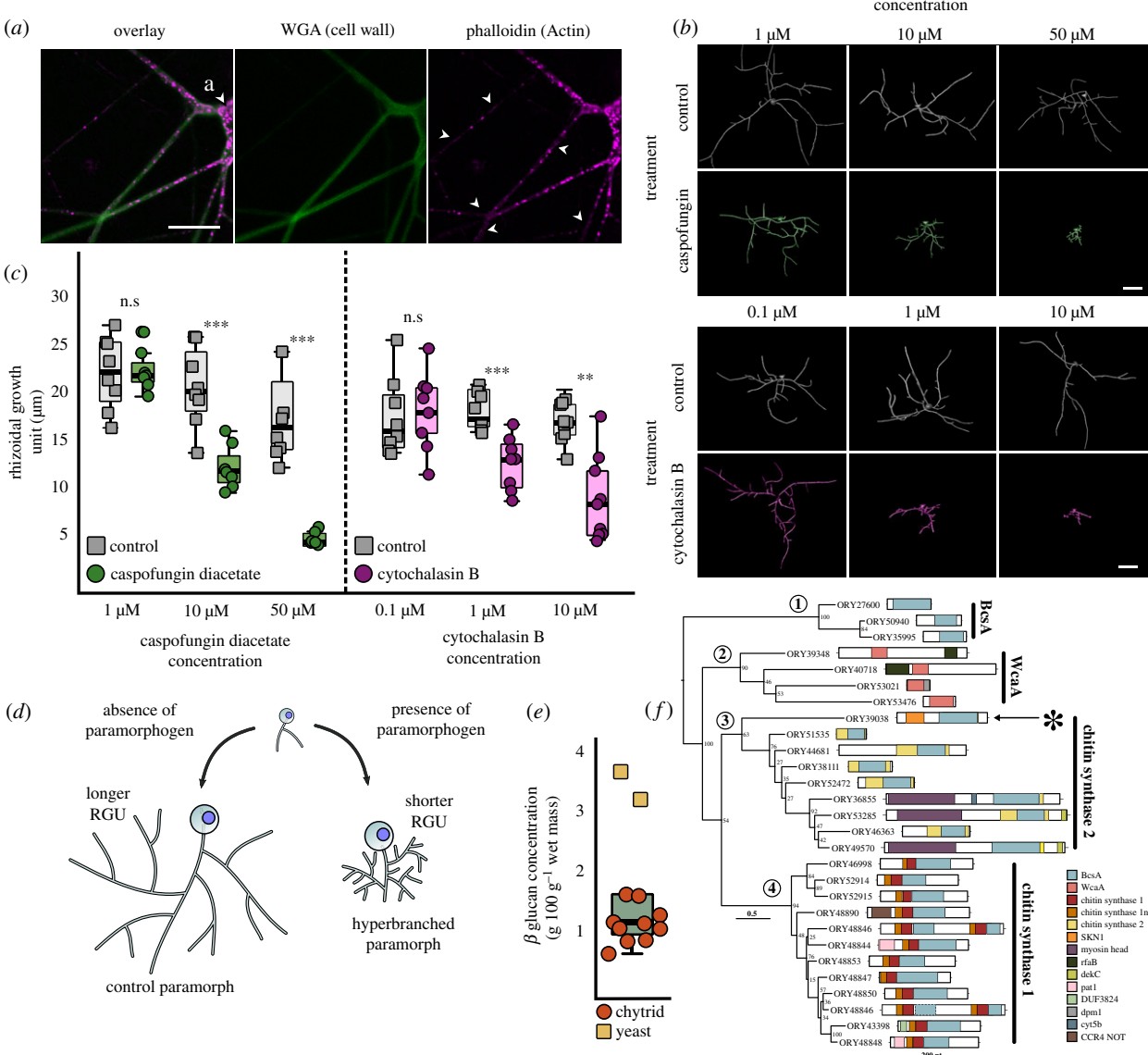

**Figure 2.** Cell wall synthesis and actin dynamics govern rhizoid branching. (a) Fluorescent labelling of cell wall and actin structures in 24 h *R. globosum* rhizoids. The cell wall and actin patches were found throughout the rhizoid. Arrowheads in the actin channel indicate putative actin cables. WGA = conjugated wheat germ agglutinin. Scale bar = 10 µm. (b) Representative three-dimensional reconstructions of 7 h *R. globosum* cells following treatment with caspofungin diacetate and cytochalasin B at stated concentrations to inhibit cell wall and actin filament biosynthesis respectively, relative to solvent only controls. Scale bar = 20 µm. (c) Application of caspofungin diacetate and cytochalasin B resulted in a concentration-dependent decrease in the rhizoidal growth unit, resulting in atypical hyperbranched rhizoids ($n \sim 9$). n.s $p > 0.05$ (not significant), *$p < 0.05$, **$p < 0.01$, ***$p < 0.001$. This differential growth is diagrammatically summarized in (d). (e) β-glucan concentration of *R. globosum* ($n = 10$) relative to a baker's yeast control ($n = 2$). (f) Maximum-likelihood phylogeny of GT2 domains (BcsA and WcaA domains) within the *R. globosum* genome (midpoint rooting). Full architecture of each protein is shown. Asterisk indicates the putative glucan synthesis protein ORY39038 containing a putative SKN1 domain.

*R. globosum* showed that they are present (figure 2e), with 58.3 ± 7.6% β-glucans and 41.6 ± 7.6% α-glucans of total glucans.

To identify alternative putative β-glucan synthesis genes in *R. globosum* JEL800, we surveyed the genome and focused on GT2 encoding genes, which include typical glucan synthases in fungi. A total of 28 GT2 domains were found within 27 genes (figure 2f). Of these genes, 20 contained putative chitin synthase domains and many contained additional domains involved in transcriptional regulation. Nine encode chitin synthase 2 family proteins and 11 encode chitin synthase 1 family proteins (with two GT2 domains in ORY48846). No obvious genes for β-1,3-glucan or β-1,6-glucan synthases were found within the genome. However, the chitin synthase 2 gene ORY39038 included a putative SKN1 domain (figure 2f), which has been implicated

in β-1,6-glucan synthesis in the ascomycete yeasts *Saccharomyces cerevisiae* [25] and *Candida albicans* [26]. These results indicate a yet uncharacterized β-glucan-dependent cell wall production process in chytrids (also targeted by caspofungin) that is not currently apparent using gene/genome level assessment and warrants further study.

## (c) Rhizoids undergo adaptive development in response to carbon starvation

To examine whether chytrids are capable of modifying rhizoid development in response to changes in resource availability, we exposed *R. globosum* to carbon starvation (i.e. development in the absence of exogenous carbon). When provided with 10 mM N-acetylglucosamine (NAG) as an exogenous carbon

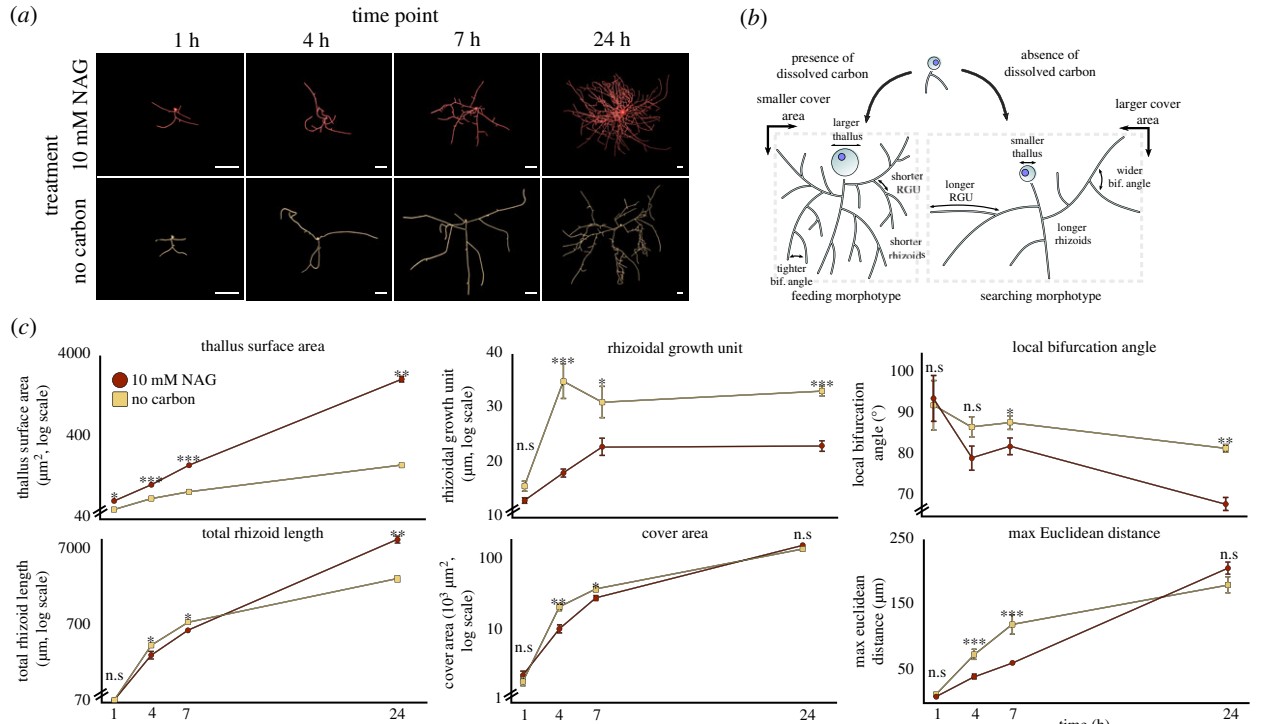

**Figure 3.** Chytrids are capable of adaptive rhizoid development under carbon starvation. (a) Representative three-dimensional reconstructions of *R. globosum* cells grown under carbon-replete or carbon-depleted conditions at different time points. Scale bar = 20 μm. When exposed to carbon starvation, chytrids are capable of differential adaptive growth to produce a searching phenotype. This differential growth is summarized in (b). (c) Differential growth trajectories of major morphometric traits between *R. globosum* cells (n ~9, mean ± s.e.m.) grown under carbon-replete and carbon-depleted conditions over time. n.s $p > 0.05$ (not significant), $*p < 0.05$, $**p < 0.01$, $***p < 0.001$.

source, the entire life cycle from zoospore to sporulation was completed and the rhizoids branched densely (electronic supplementary material, movie S8), indicative of a feeding phenotype. Carbon-starved cells did not produce zoospores and cell growth stopped after 14–16 h (electronic supplementary material, movie S9). Using only endogenous carbon (i.e. zoospore storage lipids), starved cells underwent differential rhizoid development compared to cells from the exogenous carbon-replete conditions to form an apparent adaptive searching phenotype (figure 3a,b; electronic supplementary material, table S4, movie S10). Under carbon starvation, *R. globosum* invested less in thallus growth than in carbon replete conditions and developed longer rhizoids with a greater maximum Euclidean distance (figure 3c). Carbon-starved cells were also less branched, had wider bifurcation angles and subsequently covered a larger surface area. These morphological changes in response to exogenous carbon starvation (summarized in figure 3b) suggest that individual chytrid cells are capable of differential reallocation of resources away from reproduction (i.e. the production of the zoosporangium) and towards an extended modified rhizoidal structure indicative of a resource searching phenotype.

### (d) Rhizoids spatially differentiate in response to patchy resource environments

Rhizoid growth of single cells growing on chitin microbeads was quantified as experimental POM (figure 4a,b; electronic supplementary material, movie S11). Initially, rhizoids grew along the outer surface of the bead and were probably used primarily for anchorage to the substrate. Scanning electron microscopy (SEM) showed that the rhizoids growing

externally on the chitin particle formed grooves on the bead parallel to the rhizoid axis (electronic supplementary material, figure S1f,g), suggesting extracellular enzymatic chitin degradation by the rhizoid on the outer surface. Penetration of the bead occurred during the later stages of particle colonization (figure 4a; electronic supplementary material, movie S12). Branching inside the bead emanated from 'pioneer' rhizoids that penetrated the particle (figure 4c).

Given the previous results of the searching rhizoid development in response to carbon starvation, a patchy resource environment was created using the chitin microbeads randomly distributed around individual developing cells in otherwise carbon-free media to investigate how encountering POM affected rhizoid morphology (figure 4d). We observed spatial differentiation of single-cell rhizoid systems in association with POM contact. Particle-associated rhizoids were shorter than rhizoids not in particle contact, were more branched (i.e. lower RGU), had a shorter maximum Euclidean distance and covered a smaller area (figure 4e). These rhizoid morphometrics closely resembled the feeding and searching modifications of the cells grown under carbon-replete and carbon-depleted conditions previously discussed (figures 4f and 3b) but instead are displayed simultaneously with spatial regulation in individual cells linked to POM-associated and non-associated rhizoids, respectively.

## 4. Discussion

Our results provide new insights into the developmental cell biology of chytrid fungi and highlight similarities between the organization of anucleate rhizoids and multicellular hyphae. The fundamental patterns of rhizoid morphogenesis

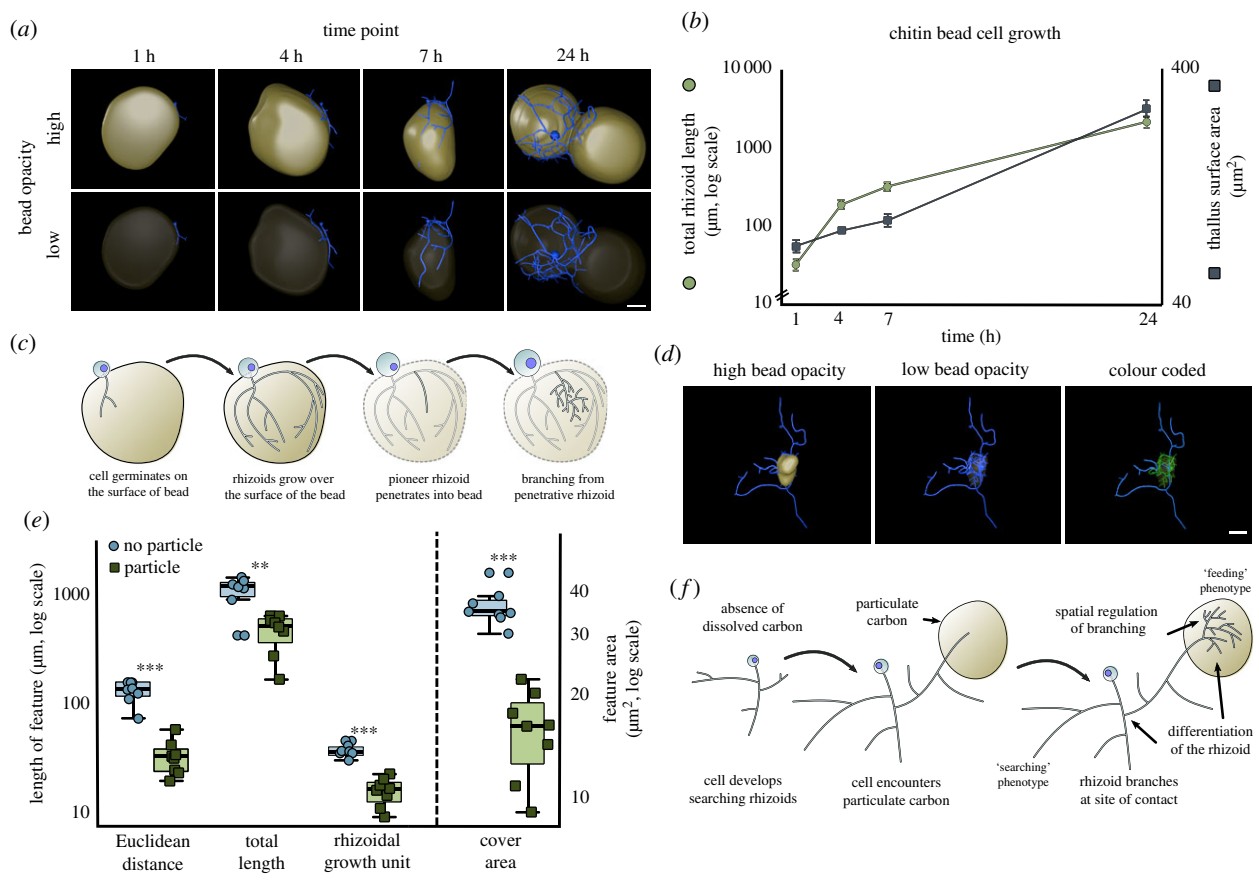

**Figure 4.** Rhizoids associated with heterogenous particulate carbon exhibit spatial differentiation. (*a*) Representative three-dimensional reconstructions of *R. globosum* cells (blue) growing on chitin microbeads (beige) at different time points. Scale bar = 20 μm. (*b*) Growth trajectories for total rhizoid length and thallus surface area for *R. globosum* cells growing on chitin microbeads (*n* ~9, mean ± s.e.m.). (*c*) Diagrammatic summary of *R. globosum* rhizoid development on chitin microbeads. (*d*) Representative three-dimensional reconstruction of a 24 h searching *R. globosum* cell (blue) that has encountered a chitin microbead (beige). The colour coded panel shows parts of the rhizoid system in contact (green) and not in contact (blue) with the microbead. Scale bar = 20 μm. (*e*) Comparison of rhizoids in contact or not in contact with the chitin microbead (*n* = 8). (*f*) Diagrammatic summary of spatial differentiation in a starved, searching rhizoid that has encountered a particulate carbon patch.

that we report here for a unicellular non-hyphal fungus are comparable to those previously recorded for hyphal fungi (figure 5*a*) [22]. Trinci [22] assessed hyphal development in major fungal lineages (Ascomycota and Mucoromycota) and observed that the growth patterns of morphometric traits (HGU, total length and number of tips) were similar across the studied taxa. When the data from our study are directly compared to that of Trinci [22], we see that the hyphal growth pattern is also analogous to the rhizoids of the early-diverging unicellular Chytridiomycota (figure 5*a*). Chytrid rhizoid development in this study is also comparable to the hyphal growth rates reported by Trinci [22] (figure 5*b*, *c*), as well as the elongation rates reported by López-Franco *et al.* [27] when scaled by filament diameter (figure 5*d*).

Such similarities also extend to rhizoid branching patterns, where lateral branching dominates over apical branching. This branching pattern is also the predominant mode of hyphal branching, where apical branching is suppressed by a phenomenon termed 'apical dominance' [28]. These findings suggest that a form of apical dominance at the growing edge rhizoid tips may suppress apical branching to maintain rhizoid network integrity as in dikaryan hyphae [28,29]. Chytrid rhizoids also become less fractal towards the growing edge in terms of their overall morphology, and similar patterns of fractal organization are also observed in hyphae-based mycelial colonies

[30]. Taken together, these results show strong geometric analogies in the fundamental organization of unicellular chytrid rhizoid and multicellular hyphal morphogenesis.

Given the apparent hyphal-like properties of rhizoid development, we sought a greater understanding of the potential subcellular machinery underpinning rhizoid morphogenesis in *R. globosum*. Normal rhizoid branching was disrupted by inhibition of cell wall synthesis and actin polymerization, both of which are known to control branching and growth in hyphal biology [31–33]. These effects in *R. globosum* are similar to disruption of normal hyphal branching reported in *Aspergillus fumigatus* (Ascomycota) in the presence of caspofungin [34], and in *Neurospora crassa* (Ascomycota) in the presence of cytochalasins [35]. Recent studies have shown the presence of actin in the rhizoids of soil chytrids [36,37] and inhibition of actin in *Chytriomyces hyalinus* similarly disrupts normal rhizoid branching [36]. In this study, our quantitative characterization of cell wall and actin inhibited rhizoid paramorphs provides support that β-1,3-glucan-dependent cell wall synthesis and actin dynamics also govern branching in chytrid rhizoids as in multicellular hyphae.

We also show that rhizoid development is plastic to resource availability, with chytrid cells displaying an adaptive searching phenotype under carbon starvation. Adaptive foraging strategies are well described in multicellular hyphae [38,39], and our data support the existence of analogous strategies in

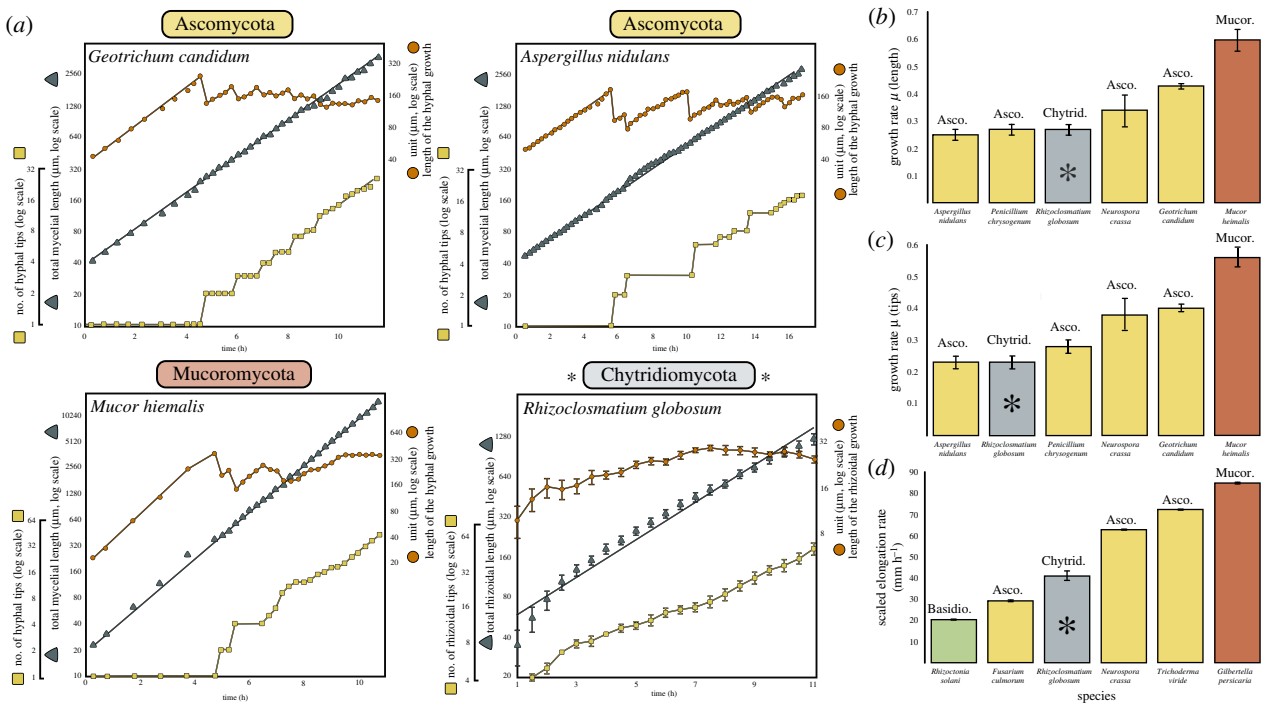

**Figure 5.** Development of chytrid rhizoids fundamentally resembles mycelial development in hyphal fungi. Comparison of rhizoid development from this study (asterisks) with other studies on hyphal fungi [22,27]. (*a*) Growth trajectories of the growth unit, total length and number of tips of rhizoids and hyphae. Data for other fungi are reproduced as new figures directly from [22]. For *R. globosum* n = 5 biological replicates. Error bars denote ± s.e.m. (*b,c*) *R. globosum* has similar growth rates regarding total length (*b*) and tip production (*c*) to hyphal fungi. Rhizoid growth rates (*μ*) calculated as increase in the total rhizoid length or tip number as in [22]. Data for Ascomycota and Mucoromycota fungi are from [22]. For *R. globosum* n = 5 biological replicates. Mean ± s.d. (*d*) *R. globosum* rhizoids when scaled by diameter have a comparable elongnation rate to hyphae. Rhizoid elongation rates (the speed at which individual rhizoid compartments extend) were quantified by measurement of extending rhizoid compartments (10 rhizoids for each biological replicate) separated by a 30 min interval on maximum intensity projected z-stacks in Fiji. Data for other fungi are from [27]. For *R. globosum* n = 5 biological replicates. Mean ± s.e.m.

rhizoidal fungi. Dense branching zones in dikaryan mycelia are known to improve colonization of trophic substrates and feeding by increasing surface area for osmotrophy, while more linear 'exploring' zones cover greater area and search for new resources [39]. Similar morphometrics are displayed by *R. globosum* exhibiting feeding and searching phenotypes, respectively. In addition, exogenous carbon starvation has also been shown to be associated with a decrease in branching in the multicellular dikaryan fungus *Aspergillus oryzae* (Ascomycota) [40]. Overall, these results highlight that adaptive search strategies are more widely spread than previously known in the Kingdom Fungi.

Finally, we report the spatial and functional differentiation of feeding and searching sections of anucleate rhizoid systems from individual cells. The simultaneous display of both rhizoid types in the same cell indicates a controlled spatial regulation of branching and differentiation of labour within single chytrid rhizoid networks. Functional division of labour is prevalently seen in multicellular mycelial fungi [38,39] including the development of specialized branching structures for increased surface area and nutrient uptake, as in the plant symbiont mycorrhiza (Glomeromycota) [41]. Our observation of similarly complex development in a unicellular chytrid suggests that multicellularity is not a prerequisite for adaptive spatial differentiation in fungi.

## 5. Conclusion

The improved understanding of chytrid rhizoid biology related to substrate attachment and feeding we present here opens the door to a greater insight into the functional ecology of chytrids

and their environmental potency. Our approach of combining live cell confocal microscopy with three-dimensional rhizoid reconstruction provides a powerful toolkit for morphometric quantification of chytrid cell development and could shed light on the biology underpinning chytrid ecological prevalence. In the future, the application of this approach to different systems could provide a detailed understanding of chytrid parasitism and host interaction, development under different nutrient regimes and degradation of diverse carbon sources.

From an evolutionary perspective, the early-diverging fungi are a critical component of the eukaryotic tree of life [42,43], including an origin of multicellularity and the establishment of the archetypal fungal hyphal form, which is responsible in part for the subsequent colonization of land by fungi, diversity expansion and interaction with plants [2]. Our cell biology focused approach advances this developing paradigm by showing that a representative unicellular, rhizoid-bearing (i.e. non-hyphal) chytrid displays hyphal-like morphogenesis, with evidence that the cell structuring mechanisms (e.g. apical dominance) underpinning chytrid rhizoid development are equivalent to reciprocal mechanisms in dikaryan fungi.

Perhaps our key discovery is that the anucleate chytrid rhizoid shows considerable developmental plasticity. *R. globosum* is able to control rhizoid morphogenesis to produce a searching form in response to carbon starvation and, from an individual cell, is capable of spatial differentiation in adaptation to patchy substrate availability indicating functional division of labour. The potential for convergent evolution aside, we propose by parsimony from the presence of analogous complex cell developmental features in an extant representative chytrid

and dikaryan fungi that adaptive rhizoids are a shared feature of their most recent common ancestor.

Data accessibility. All data that support the findings of this study are included in the electronic supplementary material of this paper.

Authors' contributions. D.L. and M.C. conceived the study. D.L. conducted the laboratory work and data analysis. N.C. analysed the *R. globosum* JEL800 genome. G.W. provided support with microscopy. M.C. secured the funding. D.L. and M.C. critically assessed and interpreted the findings. D.L and M.C. wrote the manuscript, with the help of N.C. and G.W.

Competing interests. The authors declare no competing interests.

Funding. D.L. is supported by an EnvEast Doctoral Training Partnership (DTP) PhD studentship funded from the UK Natural Environment Research Council (NERC grant no. NE/L002582/1). M.C. is supported by the European Research Council (ERC) (MYCO-CARB project; ERC grant agreement no. 772584). N.C. is supported by NERC (Marine-DNA project; NERC grant no. NE/N006151/1). G.W. is supported by an MBA Senior Research Fellowship.

Acknowledgements. The authors would like to thank Glenn Harper, Alex Strachan and the team at the Plymouth Electron Microscopy Centre (PEMC) for their assistance. We are indebted to Joyce Longcore (University of Maine) for providing *R. globosum* JEL800 from her chytrid culture collection (now curated by the Collection of Zoosporic Eufungi at the University of Michigan).

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
