## [Reviewer comments · Proceedings of the Royal Society B: Biological Sciences]

Review History

RSPB-2020-0433.R0 (Original submission)

Review form: Reviewer 1

Recommendation

Major revision is needed (please make suggestions in comments)

Scientific importance: Is the manuscript an original and important contribution to its field?

Good

General interest: Is the paper of sufficient general interest?

Acceptable

Quality of the paper: Is the overall quality of the paper suitable?

Good

Is the length of the paper justified?

Yes

Should the paper be seen by a specialist statistical reviewer?

No

Do you have any concerns about statistical analyses in this paper? If so, please specify them explicitly in your report.

No

It is a condition of publication that authors make their supporting data, code and materials available - either as supplementary material or hosted in an external repository. Please rate, if applicable, the supporting data on the following criteria.

Is it accessible?

Yes

Is it clear?

Yes

Is it adequate?

Yes

Do you have any ethical concerns with this paper?

No

Comments to the Author

The MS investigated the chytrid rhizoid development with several approaches, including 3D/4D confocal microscopy approach in combination with the application of neuron tracing software to 3D reconstruct developing cell. The 3D/4D confocal microscopy approach is quite fascinating with beautiful images and movies. All data are valuable, and most of the analyses seem to be proper. This paper will surely enrich the knowledge of chytrid biology, ecology and evolution.

I found the study has a great potential for future applications, such as examining the ability of chytrids to decompose new substrates (e.g. microplastics). In addition, it is very interesting to investigate host-parasite interactions using parasitic chytrids.

One of the main messages is that the rhizoid development of chytrid has similarities with hyphae in dikaryan fungi. Rather, I think the study showed us the new tool to investigate substrate utilization by chytrid clearly and quantitatively. Especially, the last part, plasticity of rhizoid and the way how to utilize chitin beads, was quite interesting for me, because with this tool we might be able to understand POM utilization (decomposition) quantitatively in aquatic ecosystems. It would be very useful for future studies if the authors can calculate the parameters, such as zoospore searching time, development time of sporangia, growth rate of fungi (biomass accumulation rate etc.), and decomposition rate of organic matters (decreasing rate of chitin etc.) etc. under different conditions. At least, the authors might able to calculate development time of sporangia under different conditions, which might be useful as Bruning (1991) showed in his several series of papers using parasitic chytrid.

Review form: Reviewer 2 (Stephen D. Harris)

Recommendation

Accept with minor revision (please list in comments)

Scientific importance: Is the manuscript an original and important contribution to its field?

Excellent

General interest: Is the paper of sufficient general interest?

Excellent

Quality of the paper: Is the overall quality of the paper suitable?

Excellent

Is the length of the paper justified?

Yes

Should the paper be seen by a specialist statistical reviewer?

No

Do you have any concerns about statistical analyses in this paper? If so, please specify them explicitly in your report.

No

It is a condition of publication that authors make their supporting data, code and materials available - either as supplementary material or hosted in an external repository. Please rate, if applicable, the supporting data on the following criteria.

Is it accessible?

N/A

Is it clear?

N/A

Is it adequate?

N/A

Do you have any ethical concerns with this paper?

No

Comments to the Author

In my view, this manuscript represents the first systematic attempt to test the hypothesis that the ancestry of fungal hyphae can be traced to rhizoids. Although highly polarized hyphae are one of the hallmark features of filamentous fungi, the evolutionary basis for their origins has remained enigmatic. Rhizoids formed by chytrids resemble the overall morphology of hyphae, which led to speculation that they could have served as a precursor. To test this idea, Laundon et al. adopt a morphometric approach that leverages 3D/4D confocal microscopy and neurone tracing tools to generate 3D reconstructions of rhizoids. They examine the affects of actin and cell wall perturbations on rhizoid morphology. Lastly, they test the effects of carbon starvation and patchy resources on morphology. To large extent, this study is built on the seminal work of Trinci from the 1970s that characterized the fungal duplication cycle. Despite the absence of nuclei in rhizoids, results from this study show a remarkable resemblance to hyphae in terms of growth (i.e., the rhizopod growth unit) and branching. Moreover, the role of actin filaments and beta-glucan synthesis in maintenance of normal rhizoid morphology is similar too that of hyphae. Finally, like hyphae, rhizoids are able to switch between morphologically distinct feeder and forager states in response to changing carbon availability.

The results presented in this manuscript are novel and significant. They establish a baseline for further studies to investigate whether rhizoid and hyphae share a common molecular origin. The results also raise several intriguing questions about the regulation of rhizoid growth and

branching in the absence of nuclei.

I have the following relatively minor suggestions for the authors to consider;

1. Line 63. Based on the image provided in Fig. 1A, it could be interpreted that aseptate hyphae are ancestral, and that rhizoids represent a chytrid-specific adaptation. Perhaps they could use a colour code or some other approach to represent the fraction of species/genera in each clade that form aseptate hyphae? In Fig. 1A, it might also help if the authors further clarified the difference between rhizoids and rhizomycelia.

Line 162. Given the absence of septa, it is not clear what the authors mean by the term rhizoid compartments (presumably the distance between branch points). In Fig. 1F, the circles could be interpreted as additional thalli.

Line 175 and Fig. 2. The resolution of the images shown in Fig. 2A could be improved. For example, higher magnification images would be helpful in distinguishing different types of actin structures (patches vs. cables).

Line 180. Should be FKS1.

Line 182. Because the authors raise the possibility of an unknown mechanism of beta-glucan production, an additional negative control (i.e., no beta-glucan) might be helpful to ensure the specificity of the assay.

Lines 252 and 633. The data shown in Supplemental Figure 8 is amongst the most important provided in the manuscript, as it clearly depicts the similar properties of growing rhizoids and hyphae. Accordingly, unless the authors are constrained by a limit to allowable figures, I would suggest that this figure be provided in the main text.

Line 593. There is no reference to panel G in the Figure Legend.

Decision letter (RSPB-2020-0433.R0)

11-Apr-2020

Dear Dr Cunliffe:

Your manuscript has now been peer reviewed and the reviews have been assessed by an Associate Editor. The reviewers' comments (not including confidential comments to the Editor) and the comments from the Associate Editor are included at the end of this email for your reference. As you will see, the reviewers and the Editors have raised some concerns with your manuscript and we would like to invite you to revise your manuscript to address them.

Research ethics:

Use of animals and field studies:

Please submit a copy of your revised paper within three weeks. If we do not hear from you within this time your manuscript will be rejected. If you are unable to meet this deadline please let us know as soon as possible, as we may be able to grant a short extension.

Best wishes,
Dr Sasha Dall
mailto: proceedingsb@royalsociety.org

Associate Editor
Board Member: 1
Comments to Author:

This is an interesting manuscript that uses cutting edge approaches to study chytrid rhizoid morphogenesis. Like the reviewers, I agree that the manuscript presents novel and significant insights into the biology of chytrid rhizoid cell biology, with both fundamental and methodological value to the field. Both reviewers have suggested some revisions to the manuscript that focus on addressing the timescales of development, improving the figures and inclusion of an additional control.

Reviewer(s)' Comments to Author:

Referee: 1

Comments to the Author(s)

The MS investigated the chytrid rhizoid development with several approaches, including 3D/4D confocal microscopy approach in combination with the application of neuron tracing software to 3D reconstruct developing cell. The 3D/4D confocal microscopy approach is quite fascinating with beautiful images and movies. All data are valuable, and most of the analyses seem to be proper. This paper will surely enrich the knowledge of chytrid biology, ecology and evolution.

I found the study has a great potential for future applications, such as examining the ability of chytrids to decompose new substrates (e.g. microplastics). In addition, it is very interesting to investigate host-parasite interactions using parasitic chytrids.

One of the main messages is that the rhizoid development of chytrid has similarities with hyphae in dikaryan fungi. Rather, I think the study showed us the new tool to investigate substrate utilization by chytrid clearly and quantitatively. Especially, the last part, plasticity of rhizoid and the way how to utilize chitin beads, was quite interesting for me, because with this tool we might be able to understand POM utilization (decomposition) quantitatively in aquatic ecosystems. It would be very useful for future studies if the authors can calculate the parameters, such as zoospore searching time, development time of sporangia, growth rate of fungi (biomass accumulation rate etc.), and decomposition rate of organic matters (decreasing rate of chitin etc.) etc. under different conditions. At least, the authors might be able to calculate development time of sporangia under different conditions, which might be useful as Bruning (1991) showed in his several series of papers using parasitic chytrid.

Referee: 2

Comments to the Author(s)

In my view, this manuscript represents the first systematic attempt to test the hypothesis that the ancestry of fungal hyphae can be traced to rhizoids. Although highly polarized hyphae are one of the hallmark features of filamentous fungi, the evolutionary basis for their origins has remained enigmatic. Rhizoids formed by chytrids resemble the overall morphology of hyphae, which led to speculation that they could have served as a precursor. To test this idea, Laundon et al. adopt a morphometric approach that leverages 3D/4D confocal microscopy and neurone tracing tools to generate 3D reconstructions of rhizoids. They examine the effects of actin and cell wall perturbations on rhizoid morphology. Lastly, they test the effects of carbon starvation and patchy resources on morphology. To large extent, this study is built on the seminal work of Trinci from the 1970s that characterized the fungal duplication cycle. Despite the absence of nuclei in rhizoids, results from this study show a remarkable resemblance to hyphae in terms of growth (i.e., the rhizopod growth unit) and branching. Moreover, the role of actin filaments and beta-glucan synthesis in maintenance of normal rhizoid morphology is similar too that of hyphae. Finally, like hyphae, rhizoids are able to switch between morphologically distinct feeder and forager states in response to changing carbon availability.

The results presented in this manuscript are novel and significant. They establish a baseline for further studies to investigate whether rhizoid and hyphae share a common molecular origin. The results also raise several intriguing questions about the regulation of rhizoid growth and branching in the absence of nuclei.

I have the following relatively minor suggestions for the authors to consider;

1. Line 63. Based on the image provided in Fig. 1A, it could be interpreted that aseptate hyphae are ancestral, and that rhizoids represent a chytrid-specific adaptation. Perhaps they could use a colour code or some other approach to represent the fraction of species/genera in each clade that form aseptate hyphae? In Fig. 1A, it might also help if the authors further clarified the difference between rhizoids and rhizomycelia.

Line 162. Given the absence of septa, it is not clear what the authors mean by the term rhizoid compartments (presumably the distance between branch points). In Fig. 1F, the circles could be interpreted as additional thalli.

Line 175 and Fig. 2. The resolution of the images shown in Fig. 2A could be improved. For example, higher magnification images would be helpful in distinguishing different types of actin structures (patches vs. cables).

Line 180. Should be FKS1.

Line 182. Because the authors raise the possibility of an unknown mechanism of beta-glucan production, an additional negative control (i.e., no beta-glucan) might be helpful to ensure the specificity of the assay.

Lines 252 and 633. The data shown in Supplemental Figure 8 is amongst the most important provided in the manuscript, as it clearly depicts the similar properties of growing rhizoids and hyphae. Accordingly, unless the authors are constrained by a limit to allowable figures, I would suggest that this figure be provided in the main text.

Line 593. There is no reference to panel G in the Figure Legend.

Author's Response to Decision Letter for (RSPB-2020-0433.R0)

See Appendix A.

RSPB-2020-0433.R1 (Revision)

Review form: Reviewer 1

Recommendation

Accept with minor revision (please list in comments)

Scientific importance: Is the manuscript an original and important contribution to its field?

Excellent

General interest: Is the paper of sufficient general interest?

Good

Quality of the paper: Is the overall quality of the paper suitable?

Excellent

Is the length of the paper justified?

Yes

Should the paper be seen by a specialist statistical reviewer?

No

Do you have any concerns about statistical analyses in this paper? If so, please specify them explicitly in your report.

No

It is a condition of publication that authors make their supporting data, code and materials available - either as supplementary material or hosted in an external repository. Please rate, if applicable, the supporting data on the following criteria.

Is it accessible?

Yes

Is it clear?

Yes

Is it adequate?

Yes

Do you have any ethical concerns with this paper?

No

Comments to the Author

The authors improved the MS in excellent way with additional analysis of growth comparison. I really like Figure 5 and am amazed that chytrid can grow relatively fast compared to other higher fungi. (Besides, I am also interested in why/how Mucoromycota grows so fast!) The growth comparison and Figure 5 were only explained in the discussion. I like to know in details how the growth rate was calculated. Especially, the difference between scaled elongation

rate and growth rate (length) was not clear for me. The error bars in each figure need explanation as well (variations among rhizoid, replicates, or experiments?). If possible, please analyze statistically whether the growth rates were significantly different among species (e.g. ANOVA). Those explanations can be added in figure legend or text, but it is also possible to put those in the main results and explain calculations in details in methods.

Decision letter (RSPB-2020-0433.R1)

09-May-2020

Dear Dr Cunliffe:

Your manuscript has now been peer reviewed and the reviews have been assessed by an Associate Editor. The reviewers' comments (not including confidential comments to the Editor) and the comments from the Associate Editor are included at the end of this email for your reference. As you will see, the reviewers and the Editors have raised some concerns with your manuscript and we would like to invite you to revise your manuscript to address them.

Research ethics:

Use of animals and field studies:

Please submit a copy of your revised paper within three weeks. If we do not hear from you within this time your manuscript will be rejected. If you are unable to meet this deadline please let us know as soon as possible, as we may be able to grant a short extension.

Best wishes,
Dr Sasha Dall
Editor, Proceedings B
mailto: proceedingsb@royalsociety.org

Associate Editor
Board Member: 1
Comments to Author:

The authors have comprehensively addressed the reviewer and AE comments in the revised manuscript. However, the reviewer has requested further clarification and explanation of how the growth data has been analysed (see specific comments below) that need to be addressed prior to acceptance of the manuscript.

Reviewer(s)' Comments to Author:

Referee: 1

Comments to the Author(s)

The authors improved the MS in excellent way with additional analysis of growth comparison. I really like Figure 5 and am amazed that chytrid can grow relatively fast compared to other higher fungi. (Besides, I am also interested in why/how Mucoromycota grows so fast!)

The growth comparison and Figure 5 were only explained in the discussion. I like to know in details how the growth rate was calculated. Especially, the difference between scaled elongation rate and growth rate (length) was not clear for me. The error bars in each figure need explanation as well (variations among rhizoid, replicates, or experiments?). If possible, please analyze statistically whether the growth rates were significantly different among species (e.g. ANOVA). Those explanations can be added in figure legend or text, but it is also possible to put those in the main results and explain calculations in details in methods.

Author's Response to Decision Letter for (RSPB-2020-0433.R1)

See Appendix B.

RSPB-2020-0433.R2 (Revision)

Review form: Reviewer 1 (Maiko Kagami)

Recommendation

Accept as is

Scientific importance: Is the manuscript an original and important contribution to its field?

Good

General interest: Is the paper of sufficient general interest?

Good

Quality of the paper: Is the overall quality of the paper suitable?

Good

Is the length of the paper justified?

Yes

Should the paper be seen by a specialist statistical reviewer?

Yes

Do you have any concerns about statistical analyses in this paper? If so, please specify them explicitly in your report.

No

It is a condition of publication that authors make their supporting data, code and materials available - either as supplementary material or hosted in an external repository. Please rate, if applicable, the supporting data on the following criteria.

Is it accessible?

Yes

Is it clear?

Yes

Is it adequate?

Yes

Do you have any ethical concerns with this paper?

No

Comments to the Author

The MS has been improved and I like the MS very much. Congratulation for the wonderful work with beautiful images. I am sure the paper will be cited in many different fields.

Decision letter (RSPB-2020-0433.R2)

15-May-2020

Dear Dr Cunliffe

I am pleased to inform you that your manuscript entitled "Chytrid rhizoid morphogenesis resembles hyphal development in multicellular fungi and is adaptive to resource availability" has been accepted for publication in Proceedings B.

Open Access

Paper charges

Sincerely,

Dr Sasha Dall

Associate Editor:

Board Member: 1

Comments to Author:

The authors have adequately addressed the remaining comments from the reviewer.

Appendix A

Response to Referees and Manuscript with Track Changes

Associate Editor

This is an interesting manuscript that uses cutting edge approaches to study chytrid rhizoid morphogenesis. Like the reviewers, I agree that the manuscript presents novel and significant insights into the biology of chytrid rhizoid cell biology, with both fundamental and methodological value to the field. Both reviewers have suggested some revisions to the manuscript that focus on addressing the timescales of development, improving the figures and inclusion of an additional control.

We welcome the positive comments from the Associate Editor. The revised manuscript includes a new figure (Figure 5) and text (L 257-266) on rhizoid development and growth rate in response to Referee1. We have also made the suggested improvements to figures (Figure 1B, Figure 1F, Figure 2A and new Figure 5) and included details on the additional controls requested by Referee 2 (L 132-135).

Referee 1

The MS investigated the chytrid rhizoid development with several approaches, including 3D/4D confocal microscopy approach in combination with the application of neuron tracing software to 3D reconstruct developing cell. The 3D/4D confocal microscopy approach is quite fascinating with beautiful images and movies. All data are valuable, and most of the analyses seem to be proper. This paper will surely enrich the knowledge of chytrid biology, ecology and evolution. I found the study has a great potential for future applications, such as examining the ability of chytrids to decompose new substrates (e.g. microplastics). In addition, it is very interesting to investigate host-parasite interactions using parasitic chytrids.

We thank Referee 1 for these encouraging and supportive comments.

One of the main messages is that the rhizoid development of chytrid has similarities with hyphae in dikaryan fungi. Rather, I think the study showed us the new tool to investigate substrate utilization by chytrid clearly and quantitatively. Especially, the last part, plasticity of rhizoid and the way how to utilize chitin beads, was quite interesting for me, because with this tool we might be able to understand POM utilization (decomposition) quantitatively in aquatic ecosystems. It would be very useful for future studies if the authors can calculate the parameters, such as zoospore searching time, development time of sporangia, growth rate of fungi (biomass accumulation rate etc.), and decomposition rate of organic matters (decreasing rate of chitin etc.) etc. under different conditions. At least, the authors might able to calculate development time of sporangia under different conditions, which might be useful as Bruning (1991) showed in his several series of papers using parasitic chytrid.

We welcome the suggestions from Referee 1 for future studies using the approach that we have developed and agree that there are many potential applications. In the revised manuscript, we have written additional text on potential future studies (L 318-324). We have also performed further development analysis of the rhizoids placed in the context of the timescales of hyphae development and growth rate/biomass accumulation (new Figure 5 B-D; L 257-266).

Referee 2

In my view, this manuscript represents the first systematic attempt to test the

hypothesis that the ancestry of fungal hyphae can be traced to rhizoids. Although highly polarized hyphae are one of the hallmark features of filamentous fungi, the evolutionary basis for their origins has remained enigmatic. Rhizoids formed by chytrids resemble the overall morphology of hyphae, which led to speculation that they could have served as a precursor. To test this idea, Laundon et al. adopt a morphometric approach that leverages 3D/4D confocal microscopy and neurone tracing tools to generate 3D reconstructions of rhizoids. They examine the effects of actin and cell wall perturbations on rhizoid morphology. Lastly, they test the effects of carbon starvation and patchy resources on morphology. To large extent, this study is built on the seminal work of Trinci from the 1970s that characterized the fungal duplication cycle. Despite the absence of nuclei in rhizoids, results from this study show a remarkable resemblance to hyphae in terms of growth (i.e., the rhizopod growth unit) and branching. Moreover, the role of actin filaments and beta-glucan synthesis in maintenance of normal rhizoid morphology is similar too that of hyphae. Finally, like hyphae, rhizoids are able to switch between morphologically distinct feeder and forager states in response to changing carbon availability. The results presented in this manuscript are novel and significant. They establish a baseline for further studies to investigate whether rhizoid and hyphae share a common molecular origin. The results also raise several intriguing questions about the regulation of rhizoid growth and branching in the absence of nuclei.

As with the Associate Editor and Referee 1, we welcome these encouraging and insightful comments from Referee 2.

I have the following relatively minor suggestions for the authors to consider;

Line 63. Based on the image provided in Fig. 1A, it could be interpreted that aseptate hyphae are ancestral, and that rhizoids represent a chytrid-specific adaptation. Perhaps they could use a colour code or some other approach to represent the fraction of species/genera in each clade that form aseptate hyphae? In Fig. 1A, it might also help if the authors further clarified the difference between rhizoids and rhizomycelia.

We have revised Figure 1 to give an estimation of the fraction of taxa that form aseptate hyphae. We have also further clarified the difference between rhizoids and rhizomycelia (L 496-498).

Line 162. Given the absence of septa, it is not clear what the authors mean by the term rhizoid compartments (presumably the distance between branch points).

We have revised the manuscript for clarification (now L 164, 166).

In Fig. 1F, the circles could be interpreted as additional thalli.

Figure 1F has also been modified for clarity by removing the circles.

Line 175 and Fig. 2. The resolution of the images shown in Fig. 2A could be improved. For example, higher magnification images would be helpful in distinguishing different types of actin structures (patches vs. cables).

We have replaced Figure 2A with a higher magnification image and clarified in the manuscript on actin in the rhizoids (now L 178-180).

Line 180. Should be FKS1.

This has been corrected in the new version of the manuscript (now L 185).

Line 182. Because the authors raise the possibility of an unknown mechanism of beta-glucan production, an additional negative control (i.e., no beta-glucan) might be helpful to ensure the specificity of the assay.

We used both negative and positive controls in the assay. This has been clarified in the new version of the manuscript (now L 132-135).

Lines 252 and 633. The data shown in Supplemental Figure 8 is amongst the most important provided in the manuscript, as it clearly depicts the similar properties of growing rhizoids and hyphae. Accordingly, unless the authors are constrained by a limit to allowable figures, I would suggest that this figure be provided in the main text.

As suggested, we have moved Supplementary Figure 8 to the main figures (new Figure 5) and added additional associated text (now L 257-266).

Line 593. There is no reference to panel G in the Figure Legend.

A reference to panel G has been added to the new version of the manuscript (now L 600).

Appendix B

Response to Referees and Manuscript with Track Changes

Associate Editor/Board Member 1

The authors have comprehensively addressed the reviewer and AE comments in the revised manuscript. However, the reviewer has requested further clarification and explanation of how the growth data has been analysed (see specific comments below) that need to be addressed prior to acceptance of the manuscript.

We have provided the additional clarification and explanation requested by the referee in the revised Figure 5 legend (L 566-581) and supplementary methods. Details are below.

Referee 1

The authors improved the MS in excellent way with additional analysis of growth comparison. I really like Figure 5 and am amazed that chytrid can grow relatively fast compared to other higher fungi. (Besides, I am also interested in why/how Mucoromycota grows so fast!).

We thank Referee 1 for this comment. We agree that the additional analysis suggested has improved the manuscript and adds to our study on chytrid rhizoid biology.

The growth comparison and Figure 5 were only explained in the discussion. I like to know in details how the growth rate was calculated. Especially, the difference between scaled elongation rate and growth rate (length) was not clear for me.

We had provided full details in the supplementary methods and this remains included. In the further revised version of the manuscript, we have also added explanation of growth rate calculations to the Figure 5 legend (L 572-574, 576-580).

The error bars in each figure need explanation as well (variations among rhizoid, replicates, or experiments?).

The information on the error bars and replicates has been added to the revised Figure 5 legend (L 570-571, 575-576, 581).

If possible, please analyze statistically whether the growth rates were significantly different among species (e.g. ANOVA).

It is not possible to make the statistical analysis because data from the other published studies are not available apart from what we have already presented in the manuscript. However, we do not feel this that impacts the message from our study i.e. that when scaled for size, chytrid rhizoids and multicellular fungal hyphae have comparable growth properties.

Those explanations can be added in figure legend or text, but it is also possible to put those in the main results and explain calculations in details in methods.

We have added the requested explanations to the revised figure legend (L 566-581). Information is also provided in the Supplementary Methods.